

# The Polar 5 airborne measurement of turbulence and methane fluxes during the AirMeth campaigns

Jörg Hartmann[1], Martin Gehrmann[1], Torsten Sachs[2], Katrin Kohnert[2], and Stefan Metzger[3,4]

[1]Alfred Wegener Institute, Helmholtz Centre for Polar and Marine Research, Bremerhaven, Germany
[2]GFZ German Research Centre for Geosciences, Telegrafenberg, 14473 Potsdam, Germany
[3]National Ecological Observatory Network, Battelle, 1685 38th Street, Boulder, CO 80301, USA
[4]University of Wisconsin-Madison, Dept. of Atmospheric and Oceanic Sciences, 1225 West Dayton Street, Madison, WI 53706, USA

*Correspondence to:* Jörg Hartmann (Jorg.Hartmann@AWI.de)

**Abstract.**

Low level flights over tundra wetlands in Alaska and Canada have been conducted during the AirMeth campaigns to measure turbulent methane fluxes into the atmosphere. In this paper we describe the instrumentation and new calibration procedures for the essential pressure parameters required for turbulence sensing by an aircraft that exploit suitable regular measurement

flight legs without the need for dedicated calibration patterns. We estimate the accuracy of the mean wind and the turbulence measurements. We show that airborne measurements of turbulent fluxes of methane and carbon dioxide using cavity ring down spectroscopy trace gas analysers together with established turbulence equipment achieves a relative accuracy similar to that of measurements of sensible heat flux if applied during low level flights over natural area sources. The inertial subrange of the trace gas fluctuations cannot be resolved due to insufficient high frequency precision of the analyser but since this scatter

is uncorrelated with the vertical wind velocity, the covariance and thus the flux is reproduced correctly. In the covariance spectra the -7/3 drop-off in the inertial subrange can be reproduced if sufficient data are available for averaging. For convective conditions and flight legs of several tens of kilometers we estimate the flux detection limit to about $4\,\mathrm{mg/m^2/d}$ for $\overline{w'CH_4'}$, $1.4\,\mathrm{g/m^2/d}$ for $\overline{w'CO_2'}$, and $4.2\,\mathrm{W/m^2/s}$ for the sensible heat flux.

## 1 Introduction

The atmospheric methane concentration has nearly tripled since pre-industrial times and is currently rising faster than at any time in the past two decades (Saunois et al. , 2016). Saunois, et al. suggest that this recent rise is predominatly biogenic. The contribution of arctic permafrost regions to this rise and to the global budget in general is still largely uncertain, mainly due to the unavailability of direct measurements on a regional scale. Bousquet et al. (2011) identified natural wetlands to be the main contributor to the interannual variability of the global budget. Thawing permafrost in a warming climate may further increase

the contribution of the Arctic. Advancing the knowlegde on arctic methane emission is the motivation to obtain airborne flux measurements over arctic permafrost regions.





The development of robust and precise sensors using cavity ring down spectroscopy for trace gas measurement (Baer et al. , 2002) has made direct flux measurements by eddy correlation possible. Throughout the Arctic flux measurements on tower sites have been established, but regional flux estimates for Arctic tundra areas based on extrapolations of these data currently exceed top-down estimates based on satellite data and global models by a factor of two (McGuire et al. , 2012). Measurements
by aircraft allow to extend emission studies into a regional scale and have been used to estimate methane by a budget approach (e.g. Karion et al. (2013), Cambaliza et al. (2014), Hiller et al. (2014) or by inverse modelling (Miller et al. , 2016)).

Airborne measurements of the direct flux requires the combination of a precise turbulence probe and a fast response gas analyser. Only few aircraft are capable yet to conduct methane flux measurements. Wolfe et al. (2017) used a C23 Sherpa and (Desjardins et al. , 2017) a Twin Otter to measure direct methane emission over mid-latitude agricultural areas. Over the
Alaskan North Slope Sayres et al. (2017) and Dobosy et al. (2017) flew a Diamond DA-42 for methane flux measurements. Specifically, eddy-covariance data from low-level flights can be used to create flux maps by means of direct surface projection (e.g. Mauder et al. (2008), Kohnert et al. (2017)) and data fusion (e.g. Metzger et al. (2013), Serafimovich et al. (2018)). These gridded fluxes provide unique insights into the spatial patterning of surface emissions including the location of hotspots, in a format most suitable e.g. for use with other spatial datasets and model validation.

Airborne turbulence measurements require a calibration of the inherent modification of the surrounding pressure field by the aicraft. For flux and flux map studies flight legs at constant level and constant speed are typically flown and the primary accuracy requirements are on the horizontal wind vector for footprint determination and on the vertical wind for covariances with scalars (temperature, trace gas concentration). We focus in this paper on the calibration for low level runs with approximately constant speed. As many research aircraft are used for mutiple tasks, equipment is not permanently installed and a recalibration is
necessary for each re-installation adding extra flight hour requirements per campaign. Here we show some new aspects on in-flight calibration using regular flux flight legs to find the primary calibration parameters without additional dedicated calibration patterns.

The aim of the AirMeth campaigns is to obtain measurements of methane emissions from natural area sources to close the gap between bottom-up and top-down estimates of the contribution of Arctic wetlands to the global methane budget. After a
few flights in 2011 over northern Germany and Fennoscandia, campaigns were carried out in 2012 and 2013 over the Alaskan North Slope and over the Mackenzie Delta in convective boundary layer conditions. Low level flight legs of 50 to 150 km length were combined with ascents and descents to well above the boundary layer at each end. In each of the latter campaigns some 40 hours of low level legs were flown. Figure 1 shows a typical flight pattern over the Mackenzie Delta. In this paper we describe the instrumentation, calibration procedures and the accuracies of the wind and flux measurements. Analyses of flux
patterns, footprint calculations and correlations between fluxes and surface conditions are discussed in Kohnert et al. (2017) and Serafimovich et al. (2018).



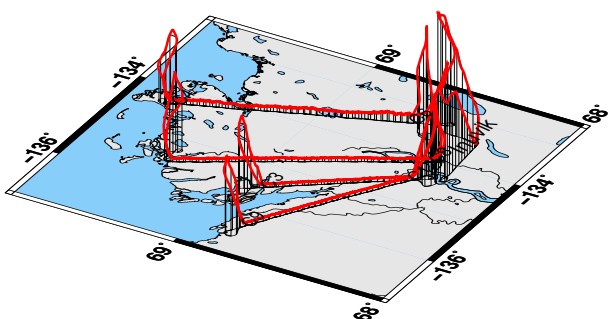

**Figure 1.** Flight path (solid red) of Polar 5 on 2013-07-20 during the AirMeth campaign illustrating a typical pattern flown with low level return track flight legs and ascents and descents for profiling the convective boundary layer.

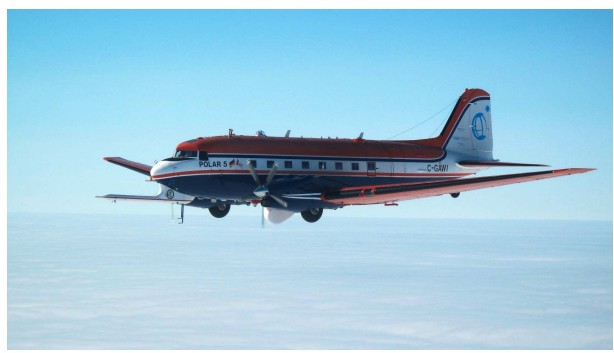

**Figure 2.** Polar 5 during a flight configured for turbulence measurements.

## 2 Aircraft and instrumentation

The airborne platform we describe in this paper is the AWI (Alfred Wegener Institute) research aircraft Polar 5, a former DC 3, converted by Basler to a turboprob aircraft and now referred to as a BT 67. Polar 5 is unpressurised, able to fly at reasonably low speed (60 m/s for low-level flux measurements, ma≈0.2) and has an endurance of 5 to 6 hours. Figure 2 shows a picture
5  of Polar 5 with the noseboom for turbulence measurements. Polar 5 is used for geosciences and atmospheric measurements and occasionally for logistics (Wesche et al. , 2016). Equipment is not permanently installed and mostly campaigns are flown with different instrumentation. Therefore the calibration coefficients and alignment offsets for the 5-hole-probe are reexamined for each reinstallation. In this paper all instrument description refer to the configuration flown in the 2012 and 2013 AirMeth campaigns.

10  ### 2.1 Turbulence probe

For turbulence measurements Polar 5 can be equipped with a noseboom carrying a 5-hole probe Type Rosemount 858. The tip of the probe is 2.9 m ahead of the tip of the fuselage. Dynamic, static and the differential pressures are measured by Rosemount





pressure transducers. For the static pressure: Rosemount 1201F2A1B1B with a precision better than 0.1 hpa between 200 and 1100 hpa, for the dynamic pressure: Rosemount 1221F2VL6B1B with a precision better than 0.02 hpa for $\pm 50$ hpa and for the flow angle differential pressures: Rosemount 1221F2VL3B1B with a precision better than 0.01 hpa for $\pm 20$ hpa. These precisions have been confirmed in laboratory calibrations with temperature variations between 0 and 20°C and during ground

recordings with the probe covered. The sensor head of the noseboom is manufactured by MessWERK (Cremer , 2008). The frequency response of the pressure transducers is sufficiently fast for atmospheric turbulence measurements as Lee (1993) found that for frequencies below 1 kHz any difference between source and measured signal cannot be attributed to the pressure sensors.

## 2.2 Position and velocity

For position, movement and attitude we use a combination of GPS and INS. The INS (inertial navigation system), a Honeywell Laseref V provides the position (longitude, latitude) at 12 Hz, the ground speed ($v_g$), true track angle ($\chi$) and true heading ($\Psi$) at 25 Hz, the pitch ($\Theta$) and roll ($\Phi$) angles at 50 Hz and the angular rates at 100 Hz. The accuracies for the angles, valid during all flight manoeuvres are gives as $0.1°$ for pitch and roll and $0.4°$ for true heading. The precision of the INS output data depends on the magnitude of flight manoeuvres (e.g. accelerations and turns). A comparison with a GPS derived direction

showed $\sigma_\Psi < 0.1°$ during a long straight and level flight. The response time of the INS is 0.02s (as given by the manual) with a delay time of about 0.01s. We found the time difference of 0.03s between INS and GPS by a cross correlation analysis of the velocity components, high-pass filtered with a cut-off at 0.001 Hz. The position and the velocity vector are supported by Novatel GPS FlexPak6. We use the Doppler-derived velocities ('Novatel bestvel') with a precision of 0.03m/s at a data rate of 1 Hz. INS and GPS are merged by complementary filtering at a frequency of 0.1 Hz.

## 2.3 Temperature and humidity

High speed temperature is recorded by an open wire Pt100 in an unheated Rosemount housing a the tip of the noseboom with a radial distance to the centre of the 5-hole probe of 16 cm and an axial distance of 35 cm. At typical measurement speed of 60 m/s the axial distance corresponds to a time lag of less than one sample at the recording frequency of 100 Hz. The effect of adiabatic heating due to the dynamic pressure has been taken into account. Humidity measurements are provided by a Vaisala

HMT-333 mounted in a Rosemount housing in a similar relation to the 5-hole probe as the fast Pt100. The HMT-333 consists of a humicap and a temperature sensor in close connection. This combination allows a correction of the humidity measurement for adiabatic heating. The calibration certificate gives the accuracies $\pm 0.4\%$ for the relative humidity and $\pm 0.1°$C for the temperature. For cross-checks a Buck-Research CR2 dew point mirror, providing highly accurate but slow absolute values, was mounted in the cabin with an inlet about 6m aft of the 5-hole probe. From 2013 on humidity was also measured in the

methane analyser. Polar 5 now also has a Licor 7200, but it was not available in the 2012 and 2013 campaigns.





## 2.4 Methane analyser

In 2011 and 2012 a Los Gatos LGR RMT 200 was rack mounted in the cabin. The RMT 200 has an internal pump enabling a slow operation mode. For flux measurements the airflow through the closed cell sensor was driven by a BOC Edwards XDS35i dry scoll pump. Outside air was taken in by a rearward facing tube 10cm above the top of the fuselage. To achieve a high flow

rate for a fast response we fed the air directly into the analyser using two filters and no air dryer. The air inlet was mounted above the cabin, 7.3m rear of the tip ot the 5-hole probe. 4.3m of stainless steel tubing with an inner diameter of 4mm (which is 54ml of volume) connected the inlet to the RMT. In 2013 an LGR-FGGA was used instead of the RMT 200. All tubing remained unchanged. In addition to $CH_4$ the FGGA also measured $CO_2$ and $H_2O$ concentrations.

## 2.5 Data recorder and sampling frequencies

Polar 5 has a state-of-the-art data aquisition and managment system ("DMS") with a high precision time based on the Precision Time Protocol according to IEEE 1588. The precision of the time stamps of all data is $\pm60$ns, the clock drift less than 1ms over 10h. Time is synchronised to the GPS. The voltage signals of the pressure transducers of the 5-hole-probe and the Pt100 temperature are digitised by 16 bit AD-converters and recorded at 100 Hz. The INS is recorded at the data rates mentioned above via a serial ARINC interface. Relative humidity at 20 Hz serial from the Vaisala interface and the CR2 data also by serial

interface at about 1 Hz. The methane data are recorded at 16 Hz in internal files by the analyser and additionally the methane concentration is fed into the DMS via an analog signal through the AD-converters to enable synchronisation.

## 3 Calibration procedures and instrumental alignment

The wind measurement by an aircraft is the usually small difference between two larger vectors: the aicraft vector with respect to Earth $V_g$ and the airflow vector with respect to the moving air $V_{\mathrm{TAS}}$:

$$V = V_g - V_{\mathrm{TAS}} \tag{1}$$

$V_g$ is given with high accuracy by the combination of INS and GPS, $V_{\mathrm{TAS}}$ is based on measurements by pressure sensors at the aircraft and transformed from the aircraft system into the local Earth system by three roations given in e.g. Lenschow and Spyers-Duran (1989), Hartmann (1990). As modifying its surrounding pressure field is the very essence of flying an aircraft heavier than air, all pressure measurements need to be calibrated to account for these modifications. Since

flying the aircraft in a wind tunnel is no option we have no other choice as to perform in-flight calibrations.

Calibration manoeuvres are described for single engine aircraft e.g. by Vellinga et al. (2013) and Mallaun et al. (2015) and for twin engine aircraft e.g. by Tjernström and Friehe (1991), Cremer (2008), and Drüe and Heinemann (2013). Typically a constant wind is assumed and speed variations are flown in box or race-track patterns for the calibration of the dynamic pressure and in level flights for the angle of attack $\alpha$. However, little attention is paid to assess the accuracy of the assumption

of a constant wind. We address that problem and describe a calibration procedure that does not need a dedicated flight pattern by exploiting a series of return-track flight legs flown for flux measurements.





## 3.1 True airspeed (TAS)

We focus on the condition of flux measurement flights, i.e. a true airspeed (TAS) of 60m/s and level flight and use the random variations in the airspeed on manually controlled flights. For an accuracy of the wind measurement better than 0.25m/s the uncertainty in the dynamic pressure needs to be smaller than 0.2hpa. As the absolute wind is virtually never known with this accuracy, we can do with the assumption of the wind changing less than this 0.25m/s during a reverse heading manoeuvre. To further overcome the uncertainty of changes in the wind field during the calibration flight, we use multiple calibration events, distributed randomly in space and time over the course of a campaign, to reduce the uncertainty of the wind assumption by $1/\sqrt{n}$, $n$ being the number of such calibration events. For example with $n$=16, we can reduce the wind uncertainty of the calibration procedure by a factor of 4. Of all flight legs during the 2013 AirMeth campaign 15 have been flown in reverse order in immediate succession. A list of these pairs of flight legs is given in Table 1. For each pair we calculate a reference ground speed

$$\overline{v_g} = \frac{1}{2}\left(\frac{v_{g1}}{\cos(\chi_1 - \Psi_1)} + \frac{v_{g2}}{\cos(\chi_2 - \Psi_2)}\right),\tag{2}$$

the indices 1 and 2 refer to the out and return legs, respectively, $(\chi - \Psi)$ is the difference angle between the true track and the true heading, as the aircraft heading deviates somewhat from the track towards the wind direction, resulting in a slightly increased TAS. Figure 3 shows a sketch illustrating the angles. In our case $(\chi - \Psi)$ is typically 2-3°, corresponding to values for $1/\cos(\chi - \Psi)$ of $\approx 1.001$, i.e. the reference ground speed is about 1‰ higher than the true groundspeed. With $\overline{v_g}$ we calculate the reference undisturbed dynamic pressure as

$$q_{gs} = \frac{1}{2}\rho\,\overline{v_g}^2\tag{3}$$

with $\rho$ being the air density. Similarly we average the indicated dynamic pressure $q_i = 0.5(q_{i1} + q_{i2})$ and use Eq.(3) to calibrate $q_i$ at the tip of the 5-hole-probe by

$$q_{gs} = c_q q_i\tag{4}$$

We find that in the range of values realised during typical low level flux runs Equation 4 is best approximated by a linear relationship, $c_q = 1.165$, shown in Figure 3. The standard deviation of the points ($q_c$) from the approximation ($1.165 q_i$) is 0.014hpa, which we take as an estimate of the calibration accuracy. The static pressure measurement can then be corrected by

$$p_s = p_{si} + q_i(1 - c_q).\tag{5}$$

## 3.2 Angle of attack alpha

At the 5-hole-probe a pressure difference results between the two holes in the vertical plane that depends on the angle of attack $\alpha$. This relation is a function of the shape of the probe and of the aerodynamical influence of the aircraft. The probe's shape has been thoroughly testet in wind tunnels e.g. (De Leo and Hagen , 1976) (Mühlbauer , 1985) and analysed theoretically (Rodi and Leon , 2012) to be expressed by a linear proportionality: $\alpha_i \sim q_\alpha/q_i$ with a proportionality constant of 12.67 and $\alpha_i$





**Table 1.** Horizontal flight legs used for the calibration of the dynamic pressure measurement and the alignment between the 5-hole-probe and the INS reference. Each line refers to one pair of return flights over the same track. The first two columns give the codes of the flight legs, further details and a full list of all flight legs is given in Kohnert et al. (2014). $l$ the averaged length, $\overline{v_g}$ the averaged ground speed, $\Delta t$ is the time difference between both legs, $\Delta \chi$ the difference in the track angle, $\Delta |U|$ the difference in the wind speed, $\Delta u$ and $\Delta v$ the differences of the horizontal wind components ($u$ positive to the East and $v$ positive to the North), $\Delta u_\perp$ and $\Delta v_\parallel$ the differences in components of the wind rotated to align with the track angle, and $\beta_r$ the remaining offset in the $\beta$-angle (10).

| leg 1 | leg 2 | $l$ | $\overline{v_g}$ | $\Delta t$ | $\chi_1$ | $\Delta \chi$ | $\Delta |U|$ | $\Delta u$ | $\Delta v$ | $\Delta u_\perp$ | $\Delta v_\parallel$ | $\beta_r$ |
|---|---|---|---|---|---|---|---|---|---|---|---|---|
| | | km | m/s | s | ° | ° | m/s | m/s | m/s | m/s | m/s | ° |
| CP50706h02 | CP50706h03 | 156.1 | 56.27 | 3814 | 96.8 | -0.5 | -0.79 | -0.79 | 0.18 | 0.10 | 0.81 | 0.02 |
| CP50711L08 | CP50711L09 | 12.0 | 64.68 | 327 | 181.5 | 0.6 | 0.20 | 0.12 | 0.68 | 0.16 | 0.62 | 0.04 |
| CP50712h01 | CP50712h02 | 90.8 | 57.77 | 2122 | 273.6 | -0.2 | 0.24 | 0.24 | 0.21 | -0.27 | 0.23 | -0.07 |
| CP50712h03 | CP50712h04 | 92.4 | 58.55 | 2145 | 93.6 | -0.0 | -0.66 | -0.65 | 0.26 | 0.23 | 0.67 | 0.06 |
| CP50719h01 | CP50719h02 | 109.8 | 59.62 | 2715 | 338.8 | -0.1 | 0.24 | 0.21 | -0.33 | -0.11 | 0.39 | -0.03 |
| CP50720h01 | CP50720h02 | 101.0 | 58.67 | 2422 | 338.9 | 0.4 | 0.29 | 0.13 | 0.05 | -0.29 | 0.09 | -0.07 |
| CP50720h03 | CP50720h04 | 118.7 | 59.49 | 2605 | 330.2 | -0.1 | -0.21 | -0.34 | 0.08 | 0.14 | -0.17 | 0.03 |
| CP50720h05 | CP50720h06 | 68.9 | 60.01 | 1636 | 324.4 | 0.1 | -0.16 | -0.27 | -0.05 | 0.16 | -0.05 | 0.04 |
| CP50720h07 | CP50720h08 | 84.3 | 60.93 | 1885 | 301.1 | 0.4 | -0.22 | -0.30 | 0.01 | 0.11 | -0.18 | 0.03 |
| CP50721L03 | CP50721L04 | 12.9 | 65.58 | 318 | 360.0 | -0.3 | 0.29 | 0.10 | 0.24 | -0.28 | -0.25 | -0.06 |
| CP50721h01 | CP50721h02 | 82.5 | 62.85 | 2727 | 179.9 | -0.4 | -0.05 | 0.07 | 0.07 | -0.07 | 0.08 | -0.02 |
| CP50721h03 | CP50721h04 | 98.7 | 65.16 | 2125 | 180.1 | -0.4 | -0.14 | 0.01 | -0.09 | -0.16 | -0.09 | -0.04 |
| CP50722h04 | CP50722h05 | 68.8 | 61.65 | 1654 | 324.4 | 0.5 | -0.09 | -0.15 | -0.35 | 0.32 | 0.21 | 0.07 |
| CP50723h02 | CP50723h03 | 86.8 | 60.34 | 2065 | 211.9 | 0.1 | -0.66 | -0.59 | -0.42 | -0.34 | -0.69 | -0.08 |
| CP50723h04 | CP50723h05 | 112.1 | 61.59 | 2458 | 209.9 | 0.0 | 0.05 | 0.10 | 0.05 | 0.02 | 0.07 | 0.01 |
| mean | | 81.3 | 62.98 | 1956 | | -0.0 | -0.11 | -0.14 | 0.04 | -0.02 | 0.12 | -0.00 |
| $\sigma$ | | | | | | 0.3 | 0.36 | 0.33 | 0.28 | 0.21 | 0.39 | 0.05 |

being the indicated, i.e. undisturbed, angle of attack, $q_\alpha$ the indicated pressure difference and $q_i$ the indicated dynamic pressure. A small dependence on the Mach number is neglected, since it is about 4 orders of magnitude smaller for the airspeed of our measurement flights. The proportionality constant is valid for a probe in an undisturbed flow, but the influence of the aircraft leads to a deviation from this number. Crawford et al. (1996) explained this deviations in terms of "lift induced upwash" in front of the aircraft. Furthermore the $\alpha$ measurement needs to be aligned with the coordinate system of the INS. This alignment may be different for each re-installation of the noseboom. Therefore an $\alpha$ calibration is typically done for each remounting of the probe and any change in the configuration of the aircraft. We combine the effects of probe shape and aircraft influence in a single calibration procedure. For the small angles that occur during straight level flights $\alpha$ depends with a very good



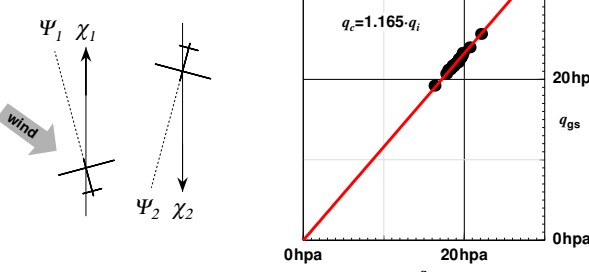

**Figure 3.** left: illustration of the angles true track $\chi$ and true heading $\Psi$ for reverse heading flights. right: dynamic pressure derived by Equation 3 versus the indicated dynamic pressure at the tip of the 5-hole-probe. Each of the 15 dots represents the average of two overpasses of the same track in revese direction. The red line is a linear regression.

approximation linearly on the pressure difference normalised by the dynamic pressure:

$$\alpha = \alpha_0 + c_\alpha \frac{q_\alpha}{q_i} \tag{6}$$

with $\alpha_0$ being the offset angle between the 5-hole-probe and the reference of the INS, and $c_\alpha$ the proportionality constant.

### 3.2.1 Dedicated calibration flight

For a calibration flight pattern we use the fact that a) with no pressure influence by the aircraft the angle of attack $\alpha$ equals the pitch angle during a straight and level flight with no vertical movement of the air and that b) for a plane with fixed aerofoil (no flap movement) $\alpha$ varies with airspeed. This is a very commonly used method for the $\alpha$-calibration. We performed three low level flight sections over water with the airspeed gradualy increasing from 50m/s to 90m/s during 5 minutes and decreasing back to 50m/s again during 5 minutes. For these data Figure 4 shows pitch versus $q_\alpha/q_i$. As the aircraft is manually controlled

during this manoeuvre and the vertical movement of the air is not constantly zero, points scatter vertically with the vertical speed of the aircraft $w_g$ and horizontally with vertical wind velocity $w$. The colour coding with $w_g$ shows that most of the scatter in explained by vertical movement of the aircraft. Typically this is being assumed to cancel on average (e.g. Mallaun et al. , 2015) and mean values over subsections are used for the calibration. This implicitly assumes a Gaussian distribution of $w$ and $w_g$.

With the quite accurate knowledge of $w_g$ we can restrict the data used for a regression to these conditions of very little vertical movement of air and aircraft and level wings, i.e.:

$$|w_g| < 0.05\text{m/s} \ \wedge \ |w| < 0.1\text{m/s} \ \wedge \ |\Phi| < 5° \tag{7}$$

and furthermore correct the pitch angle by

$$\Theta_c = \Theta - \frac{w_g}{\text{TAS}} \frac{180}{\pi} \tag{8}$$





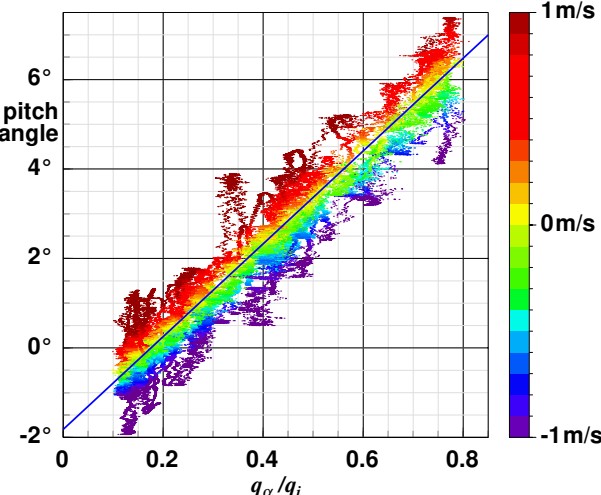

**Figure 4.** Pitch angle $\Theta$ versus alpha pressure difference normalised by the dynamic pressure $q_\alpha/q_i$. The data are from three low-level calibration flight sections over water off Barrow on 6 July 2013. Colour coded is the vertical velocity of the aircraft $w_g$ with the colour scale given in the vertical bar at the right. Plotted are the 100Hz data. The blue line represents the linear regression $\alpha = -1.822 + 10.375\frac{q_\alpha}{q_i}$.

to account for the remaining small vertical movement of the plane to arrive at

$$\alpha_0 = -1.822 \pm 0.033 \quad \text{and} \quad c_\alpha = 10.375 \pm 0.073. \tag{9}$$

Note that for our data (Eq. 7) the correction term in (8) is smaller than $0.05°$. As the vertical wind velocity, needed for the selection condition (7), is not known before the final calibration coefficients are determined, we need to run through one step

of iteration for which we use the coefficients of the most recent campaign as a first guess. The uncertainties in the regression coefficients in (9) translate into an offset uncertainty for the vertical wind velocity of $\sim 0.03$m/s and a gain uncertainty of $\sim 0.7\%$. Our value for $c_\alpha$ is close to that of Mallaun et al. (2015) who found a correction factor of 0.78 necessary for theoretical value of 12.66 to account for the aircraft influence of a Cessna Grand Caravan. A Gaussian error propagation for Eq. 6 with $q_i$=20 hpa (TAS $\approx$ 60 m/s) and $q_a$=10 hpa (vertical wind $\approx$ 1 m/s) and using the uncertainties $0.033°$ for $\alpha_0$, $0.073°$ for $c_\alpha$,

0.01 hpa for $q_\alpha$ and 0.02 hpa for $q_i$ yields an uncertainty for $\alpha$ of $0.05°$, with the dominating contribution from the uncertainty of the regression slope.

### 3.2.2 No need for calibration flight for $\alpha$

It is interesting to note that an $\alpha$ calibration is actually possible without any specific flight manoeuvre if sufficient data are available. We demonstrate this for the AirMeth campaign in 2013. We use all flight data of all days, except the 6[th] of July

2013, the day of the dedicated $\alpha$-pattern, to have an independent test. Of these 68 h of flight data (with $q_i > 10$ hpa, $\cong$ 50 m/s to ensure in-flight conditions) we select those that fulfill the conditions given in (7): vertical movement of the plane smaller than 5 cm/s, vertical wind velocity smaller than 0.1 m/s, roll angle smaller than $5°$ (absolute values for each). Roughly 0.6 % of the





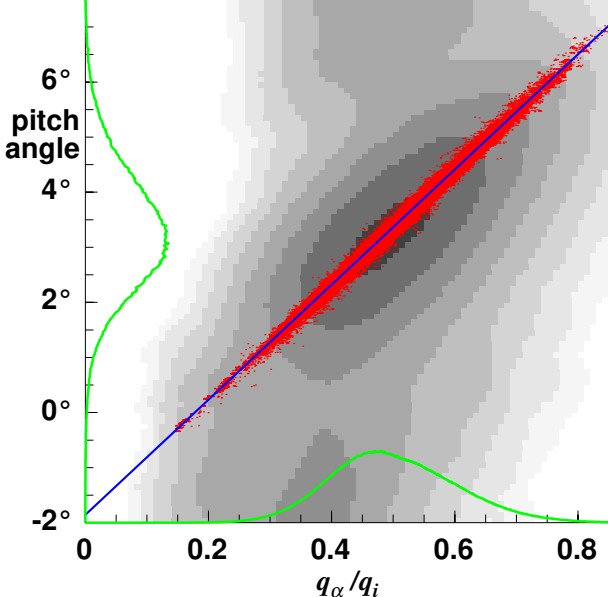

**Figure 5.** Pitch angle versus alpha pressure difference normalised by the dynamic pressure for all flights (except 6 July 2013) during the 2013 AirMeth campaign. Red dots are data that fulfill the conditions given in (7) and with correction of the pitch angle Eq. (8). Grey shading indicates the distribution of all data which includes ascents, descents, take-off and landing procedures. A logarithmic shading scale is used. Only data with $q_i < 10$hpa (corresponding to TAS=50m/s) are excluded to ensure in-flight conditions. The green lines show the normalised frequency distribution of all data of the horizontal level runs used for flux measurements. The blue line represents the regression $\alpha = -1.856 + 10.449 \frac{q_\alpha}{q_i}$.

data remain and are plotted in Figure 5 as red dots. For comparison grey shading indicates the density distribution of all 68h of data. A least squares fit of a linear relation results in $\alpha_0 = -1.856 \pm 0.016$ and $c_\alpha = 10.449 \pm 0.030$, which are within the range of uncertainty of (9) and differ from the values of the dedicated pattern by 1.8% for the offset and 0.7% for the slope. At the typical airspeed during measurements runs of 60m/s the offset corresponds to constant difference for the vertical wind speed of 3cm/s and the slope deviation translates directly to an gain difference of 0.7%. Both figures are in the range of uncertainty of the results of the dedicated flight pattern.

From Fig. 5 we can furthermore see that the pitch variation during measurement runs are nearly Gaussian distributed, while the pressure ratio $q_\alpha/q_i$ is positively skewed due to the skewness of $w$ at low level in a convective boundary layer (e.g. Hunt et al. , 1988).





### 3.3 Alignment of the 5-hole-probe and INS reference (beta-offset)

The angle of sideslip $\beta$ is measured by the 5-hole-probe via the pressure difference $q_\beta$ between the two holes in the horizontal plane. Then $\beta$ is calculated by

$$\beta = \beta_0 + c_\beta \frac{q_\beta}{q_i} \tag{10}$$

where $\beta_0$ is the alignment offset between the 5-hole-probe and the INS reference system and $c_\beta$ in analogy to $c_\alpha$ (Eq. 6) a proportionality constant. For a symmetrical sonde in a pressure field undistorted by the aircraft $c_\alpha$ and $c_\beta$ would be identical, as e.g. Mühlbauer (1985) proved in a wind tunnel. But as we include in the calibration the influence of the aircraft pressure field which is not symmetrical with respect to the longitudinal axis of the sonde, $c_\alpha$ and $c_\beta$ are different. The porportionality $c_\beta$ should not change between campaigns, but $\beta_0$ needs to be recalibrated with each remounting of the noseboom. We use

$c_\beta = 11.36$ as determined in the calibration flights of Cremer (2008) and confirmed by Drüe and Heinemann (2013). Based on the assumption that the wind be constant for the out and return flights we calculate for each pair of legs

$$\beta_r = \mathrm{atan}(0.5 \frac{u_{\perp 1} + u_{\perp 2}}{\mathrm{TAS}_1 + \mathrm{TAS}_2}) \tag{11}$$

as a residual offset for the beta angle. We then manually iterate the beta-offset $\beta_0$ such that the average over all $\beta_r$s is minimised. For the AirMeth 2013 flights we find $\beta_0 = -0.604$. Mallaun et al. (2015) pointed out that a misalignment of the $\beta$ angle

should show in a correlation between the vertical wind velocity and the roll angle, as a misaligned sonde would be tilted up- or downward and thus produce a spurious vertical wind. Following their suggestion we testet $w \sim \mathrm{TAS} \cdot \sin\Phi$ for $\Phi > 5°$ and $|w_g| < 0.1$m/s and could not find any correlation.

### 3.4 Static pressure precision

We can use the series of return track flights for an estimation of the precision of the static pressure measurement. As we have

passed during the return-flight the same location (with about $\pm 200$ m lateral deviation), we can calculate a pressure difference along the track. This difference is composed of sensor uncertainties, height variation of the aircraft and atmospheric change. The height variation is accounted for by calculating the static pressure for a reference height $h_\mathrm{ref}$ by

$$p_\mathrm{ref} = p_s + (h - h_\mathrm{ref}) \frac{p_s g}{RT} \tag{12}$$

with $p_s$ the static pressure (Eq. 5), $g$ the acceleration due to gravity, $R$ the gas constant for air and $T$ the temperature. As

reference height we define the mean over both flight legs. The atmospheric change is handled by this procedure: for each position along the track we have a

$$\Delta p = |p_\mathrm{ref2} - p_\mathrm{ref1}|, \tag{13}$$

the absolute value of the pressure difference between both passes, and in analogy a $\Delta t$, the time elapsed between both over-passes. Plotting $\Delta p$ versus $\Delta t$ shows increasing scatter with increasing $\Delta t$. A least squares fit gives at the ordinate offset at



$\Delta t$=0 the remaining uncertainties of the sensors. We find this $\Delta p_0$ to be $< 0.1$hpa. This uncertainty estimate includes the uncertainty of the direct pressure measurement as well as that due to the aircraft height based on the gps data. With this uncertainty a pressure gradient detection limit for a 100 km long leg would be 0.001hpa/km.

### 3.5 Accuracy of the horizontal wind measurement

The difference in the mean wind speed $\Delta|U|$ between out and return legs as shown in Table 1 has over all 15 pairs a mean value of 0.08m/s and a standard deviation of 0.33m/s. This supports our assumption that $\Delta|U|$ mostly results from atmospheric variation and that the calibration and measurement uncertainty rather is of the order of 0.08m/s. Rotating the wind components in an along track $v_\parallel$ and an across track $u_\perp$ component we get a mean difference in $v_\parallel$ of 0.11m/s which translates in a calibration uncertainty for the dynamic pressure of $\approx 0.09$hpa and is of similar order as the estimate given in section 3.1.

Calculating a Gaussian error propagation on the by far dominating term for the along track component

$$v_\parallel = v_{g\parallel} - \sqrt{\frac{2q_i RT}{p}} \tag{14}$$

using the uncertainties 0.1hpa for the static pressure $p$, 0.12hpa (averaging the estimates of Sections 3.1 and 3.5) for the dynamic pressure $q_i$, 0.1K for the temperature $T$, and 0.03m/s for the ground speed $v_{g\parallel}$ results in an uncertainty for $v_\parallel$ of 0.18m/s. In this estimate the uncertainty of $q_i$ clearly dominates the other contributions by about one order of magnitude. Note that this estimate

is valid for wind measurements during horizontal flight legs. The accuracy during turn manoeuvres, ascents and descents may be less. For the alignment offset between the 5-hole-probe and the INS we estimate the calibration uncertainty by the standard deviation of $\beta_r$, given in Table 1 (second last line) to be $0.05°$. Furthermore, applying the procedure described in section 3.4 to the horizontal wind components yields as uncertainty estimates for both components of 0.2 m/s, confirming the estimate in this section.

### 3.6 Methane analyser

The data aquisition system of Polar 5, DMS, and the methane analyser ran on autonomous computer system each with their individual clocks. They were synchronised by recording within the DMS an analogue output of the methane analyser. Section-wise cross correlation revealed that the analyser's clock ran typically 3.5e-5 slower than the DMS. This sychronisation was done individually for each flight leading to a timing accuracy of 0.01s between the systems.

After clock synchronisations, the time lag of the methane signal due to delay in the tubings was found by a cross-correlation analysis of the FGGA data with the vertical wind velocity for selected runs with clearly positive methane and humidity fluxes. Prior to the correlation analysis all signals were high-pass filtered with a cut-off at 0.1 Hz (corresponding to $\approx 600$ m horizontal distance at 60 m/s). The time lags for $CH_4$, $CO_2$ and $H_2O$ are 0.68s, 0.66s and 0.72s, respectively, with negligible variation between individual runs. Water vapour has a slightly larger delay due to interaction with the tubing. However, as Ibrom et al.

(2007) have shown, for referencing the methane signal to a dry mole fraction the water vapour signal needs to be treated with the same time delay as the methane signal, as the actual condition in the measurement cell are relevant. A correlation analysis



between the FGGA and Vaisala humidity signals showed a delay of 0.36s of the Vaisala signal. The time delay of the methane signal due to the tubing was confirmed by ground a test. A step change of the concentration a the inlet took 0.5s to arrive at the analysers reading.

The cell pressure in the methane analyser is maintained to 140 Torr and shows little variation during level flux measurement
runs. Desjardins et al. (2017) used a Picarro G2301-f in a Twin-Otter for flux measurements and found a weak correlation of the methane concentration with the cell pressure. We performed coherency and correlation analysis with spectral resolution and as integral statistics and could not find any correlation between pressure and the $CH_4$ signal. Also Wolfe et al. (2017) reported no pressure effect on the $CH_4$ signal from an airborne LGR analyser.

A specific, and especially arctic problem of airborne cavity ring down spectroscopy is sensor warm up. In a flux tower setup
sensors typically run continuously, but for airborne applications the instruments can only be switched on after start of the engines. Occasionally sensors could be pre heated by ground power but this was not always available. Laboratory and in-flight tests showed, that the $CH_4$ concentration reported from a cold sensor increased with cavity temperature for temperures lower than 34°C. For below-zero starting condition warm-up time was up to 45 minutes.

## 4   Accuracy of methane flux measurements

To analyse the accuracy of airborne trace gas flux measurements we estimate the flux detection limit, test the instrument precision and use a spectral analysis to compare methane fluxes with the well known behaviour of heat and moisture fluxes. We focus on the covariance at the height of the aircraft. For referencing the flux measurement to the surface level and footprint calculations we refer to Kohnert et al. (2017) and Serafimovich et al. (2018).

### 4.1   Turbulent flux detection limit

Following a method suggested by Wienhold et al. (1995) we use the cross covariance function to estimate the flux detection limit. We calculate the standard deviation for the time lag interval -200s to -50s and 50s to 200s. At a typical airspeed of 60m/s this corresponds to 3 to 12km horizontal distance. Figure 6 shows an example for a horizontal flight section on 2013-07-13.

Applying this procedure to all horizontal flight legs of the 2013 campaign with positive methane, heat and moisture fluxes and negative CO2 fluxes and averaging we get detection limits of 3.9 mg/m$^2$/d for $\overline{w'CH_4'}$, 1.4 g/m$^2$/d for $\overline{w'CO_2'}$, 4.2 W/m$^2$/s
for the sensible heat flux and 8.8 W/m$^2$/s for the latent heat flux.

Applying the Billesbach (2011) method to all 44 low level flight legs of the AirMeth 2012 North Slope campaign yields comparable flux detection limits of 4.9±1.4g/m$^2$/d 4.6±1.9W/m$^2$, 3.9±1.3W/m$^2$ and for the fluxes of methane, sensible and latent heat, respectively. The LGR RMT 200 sensor installed in 2012 did not measure $CO_2$.

### 4.2   Precision

To determine the instrumental noise level from our recordings we follow a method described by Mauder et al. (2013), based on the property of white noise being uncorrelated with the signal. Thus it shows only in the 0-lag of the autocorrelation but not



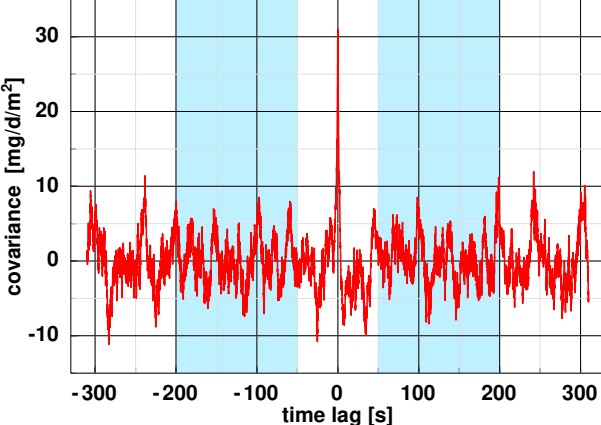

**Figure 6.** Example of the covariance function of $w'$ and $CH_4'$ versus time lag to illustrate the range used for estimation of the flux detection limit. The covariance is scaled to mg/d/m$^2$. Blue shaded areas indicate the ranges -200s to -50s and 50s to 200s over which the standard deviation has be calculated to estimate the flux detection limit. At the typical airspeed of 60m/s the range corresponds to 3 to 12 km. The figure shows data of run CP50713h02 with a methane flux of 30.9mg/m$^2$/d. The flux detection limit is 3.1mg/m$^2$/d

in futher lags. The variance of the noise error can be estimated as

$$\overline{\epsilon x^2} = C_{11}(0) - C_{11}(p \rightarrow 0)$$

where $C_{11}(0)$ is the autocorrelation of $x$ at lag 0 and $C_{11}(p \rightarrow 0)$ the autocorrelation extrapolated to lag 0. For the FGGA we get for CH$_4$ $\epsilon x = 0.0037$ppm, for CO$_2$ $\epsilon x = 0.695$ppm and for H$_2$O $\epsilon x = 34.9$ppm, all confirming the design specifications of the instrument. Applying the same proceedure to the data of vertical wind velocity and temperature we get for $w$: $\epsilon x = 0.029$m/s and for T$\epsilon x = 0.0022$K We use for $C_{11}(p \rightarrow 0)$ the lags 3-20 corresponding to 0.16s to 1.0s sampling time.

## 4.3 Spectral analysis

With the precision of $\pm 3$ppb for an integration time of 0.1s of the methane analyser we cannot expect to have spectral resolution of atmospheric fluctuations in the high frequency range that is comparable to temperature and vertical velocity. We examine power spectra (Figure 7) of a 100 km long flight leg at 50 m above ground. The measurement were taken on 2013-07-12 over the North Slope of Alaska in a convective boundary layer driven by a sensible heat flux of 70 W/m$^2$. The boundary layer height $z_i$ was 500 m. Vertical wind velocity and temperature nicely follow a -5/3 drop off over nearly 2 decades for horizontal scales smaller than the boundary layer height. The data from the FGGA contain considerable white noise, most pronounced for CO$_2$, followed by CH$_4$ and least for the water vapour measurement. All three show too much HF-noise to resolve the inertial subrange of turbulence. Similar results are shown by Wolfe et al. (2017) from low level airborne carbon flux measurements over Maryland and Virginia. Beyond about 5 Hz (corresponding to 12 m horizontal distance at the typical airspeed of 60 m/s) spectral drop off due to dampening in the tubing is visible. As $w$ scales with the boundary layer height, power at the low





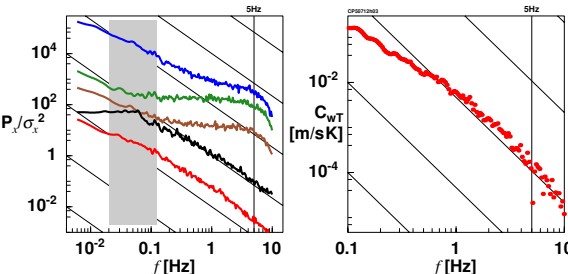

**Figure 7.** left: power spectra of the fluctuation of temperature (red), vertical wind velocity (black), $CH_4$ (brown), $CO_2$ (green) and water vapour mixing ratio (blue). The spectra are nondimensionalised by their respective variance and shifted in the plot by one decade increasingly. The sloped lines indicate a -5/3 decrease. The grey shaded area marks the scales corresponding from $5 z_i$ to $0.5 z_i$, the range of dominant transport in a convective boundary layer. Right: covariance spectrum of vertical wind velocity and temperature. The sloped lines indicate -7/3 decrease. Data are from 2013-07-12, Alaskan North Slope, measurement height above ground 50 m, boundary layer height $z_i$=500m above ground.

frequency end does not increase further while the fluctuations in all scalars continue on scales far beyond 100 times the boundary layer height since the scalar quantities rather scale with their horizontal surface structure.

In the cospectrum of $w$ and $T$ we see the expected -7/3 drop off (e.g. Kaimal et al. (1972)), as shown in Figure 7. Beyond 5 Hz shows a small drop off, however, theses scales (corresponding to 12 m horizontal resolution) contribute a negligible
amount to the covariance at the aircraft height of 50 m. The uncertainties at the low-frequency end are larger and more important for flux estimates.

Since the white noise of the trace gas analyser is uncorrelated with the vertical velocity it does not show in the covariance spectra (Figure 8). All 4 spectra are of similar shape. Although $C_{wCH_4}$ and $C_{wCO_2}$ have considerably more scatter in the high frequencies, their drop-off follows that of $C_{wT}$. Thus the turbulent vertical transport of trace gases is essentially identical to
that of other scalars in the convective boundary layer.

Uncorrelated instrumental noise should vanish, or at least reduce, if measurements are repeated under similar conditions and averaged. The statistical error then reduces proportionally to $\frac{1}{\sqrt{n}}$, $n$ being the number of independent realisations. We calculated covariance spectra for each of the 93 available low level legs of the 2013 AirMeth campaign, normalised by their covariance and averaged. In these stacked cospectra (Figure 9) the expected -7/3 drop-off is reproduced for all 4 scalar fluxes.
Again, with more scatter for the trace gases than for the water vapour or the temperature. Figure 9 shows that the instrumental noise leading to the spectral deviation in Figure 7 is uncorrelated with the vertical velocity and does not affect the covariance other than by a small increase of statistical uncertainty.





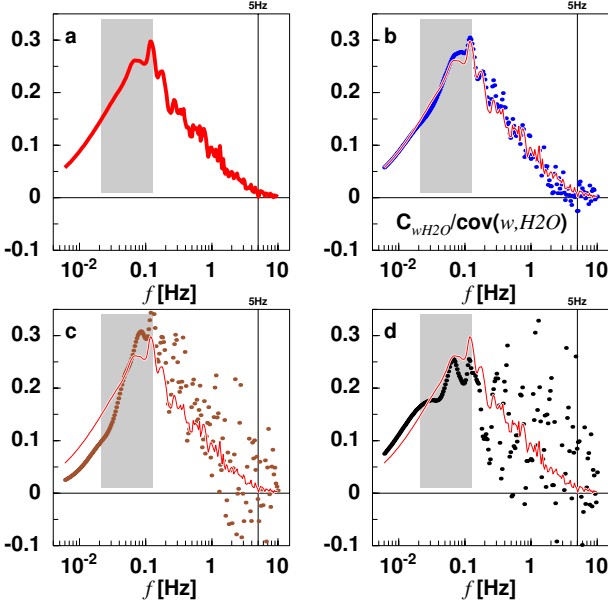

**Figure 8.** Cospectra normalised by their respective covariances. The data are from the same flight leg as in Figure 7. The grey shaded area marks the scales corresponding from $5 z_i$ to $0.5 z_i$. **a:** $C_{wT}$ sensible heat flux, red, **b:** $C_{wH2O}$ moisture flux, blue, **c:** $C_{wCH4}$ methane flux, brown and **d:** $C_{wCO2}$ flux of carbon dioxide, black. Note that normalisation by the covariance eliminates the sign. The first three fluxes are upward directed, the carbon dioxide flux is downward. For comparison $C_{wT}$ is plotted as thin red line in b),c) and d).

## 4.4 Dry mole fraction flux

We aim to determine the mass of methane being emitted from the surface per area unit and time interval. The trace gas analyser measures molecular ratios. As the atmospheric methane concentration is of a similar order as the density variation due to humidity fluctuations, the latter need to be taken in to account when referencing the measured (wet) mole fractions to a mass
flux (Webb et al. , 1980)

A direct measurement of dry mole fractions requires gas drying. However, for eddy correlation analysis a fast response of the system is very important. To keep the tubing as short as possible, we fed the outside air directly into the analyser avoiding delays by an air dryer, and account for the effect of humidity fluctuations by using fast humidity measurements. This method can even be applied in the tropics with considerably higher atmospheric humidity as Chen et al. (2010) have proven. To then
find the dry mole fraction flux two options remain:

**1:** finding for each $CH_4$ sample taken in the measurement cell the exact humidity in the very same moment. For this method either an additional humidity measurement needs to be done in the analyser cell, or a separate fast humidity measurement can be referenced into the analyser with a high temporal accuracy.

**2:** calculating a wet mole fraction flux and applying what is commonly referred to as one of two WPL-correction terms
(Webb et al. , 1980). For method **2** a separate humidity flux measurement needs to be available.





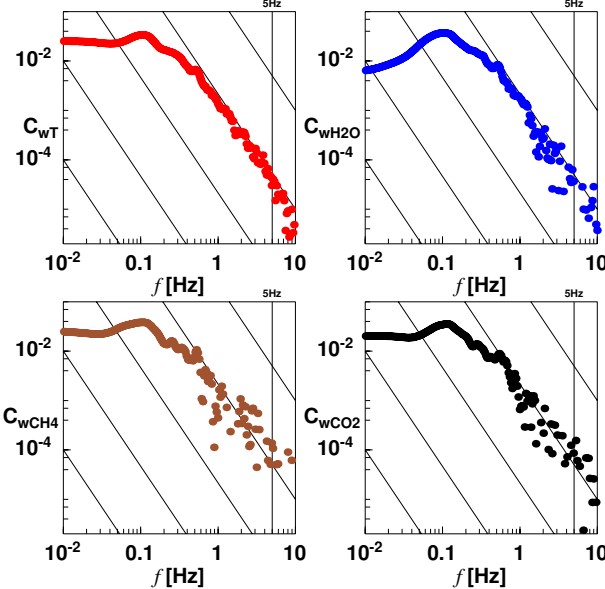

**Figure 9.** Stacked cospectra normalised by their respective covariances. The spectra are averages of 93 horizontal flight legs totalling some 6000 km distance and 28 h. **top left:** $C_{wT}$ sensible heat flux, red, **top right:** $C_{wH2O}$ moisture flux, blue, **bottom left:** $C_{wCH4}$ methane flux, brown and **bottom right:** $C_{wCO2}$ flux of carbon dioxide, black. Note that normalisation by the covariance eliminates the sign. Thin black lines show the -7/3 slope.

With the FGGA used in the 2013 AirMeth campaign the water vapour concentration is measured in the same air volume and at the same time as the trace gas concentration. Dry mole fraction can then be calculated by

$$\text{CH}_{4d} = \frac{\text{CH}_{4w}}{(1 - \text{mr}_{\text{H}_2\text{O}})} \tag{15}$$

where $\text{mr}_{\text{H}_2\text{O}}$ is the ratio of water vapour to dry air. The dry mole flux then is

$$F = \overline{w'\rho'_{\text{CH4}_d}} \tag{16}$$

We use these data to estimate differences and possible inaccuracies introduced by the above mentioned methods. We compare the dry mole fraction flux based on $\text{CH}_{4d}$ with these four different methods:

**A** based on $\text{CH}_{4w}$ plus the WPL-term calculated from the FGGA-humidity measurement.

$$F_A = \overline{w'\rho'_{\text{CH4}w}} + \frac{m_a}{m_v}\left(\frac{\overline{\rho_{CH4}}}{\overline{\rho_a}}\right)\overline{w'\rho'_{v\text{FGGA}}} \tag{17}$$

$m_a/m_v$ is the mass ratio of dry air and water vapour, $\rho_{CH4}$ and $\rho_a$ the densities of methane and dry air, respectively, and $\rho_{v\text{FGGA}}$ the water vapour density as measured by the FGGA. $F$ and $F_A$ should only be affected by numerical inaccuracies. The ratio $F_A/F$ turns out to be 0.993±0.002.





**B** based on $CH_{4w}$ plus the WPL-term taken from the Vaisala-humidity measurement.

$$F_B = \overline{w'\rho'_{\text{CH4}w}} + \frac{m_a}{m_v}\left(\frac{\overline{\rho_{CH4}}}{\overline{\rho_a}}\right)\overline{w'\rho'_{v\text{VAIS}}} \tag{18}$$

The ratio $F_B/F$ turns out to be $1.041\pm0.035$. The overestimation of 4.1% is due to the fact that the Vaisala measurement leads to a 31.2% larger humidity flux than the FGGA-measurement. However a direct comparison between averaged humidity measurements shows a good agreement. The flux difference is due to a different response behaviour of both sensors. Since in the 2012 campaign no other fast humidity measurement was available, this method had to be applied, leading to a slightly increased uncertainty of the methane flux. Assuming for 2012 a siminlar behaviour as for 2013, we roughly 4% overestimation of the methane fluxes.

**C** based on $CH_{4d\text{vais}}$ as calculated from $CH_{4w}$ and the in-cell humidity derived from the (outside) Vaisala measurement (HMT-330) referenced into the analyser cell. We calculated the mixing ratio from the relative humidity, temperature (Pt100) and the pressure and determined the time lag to the humidity measurement of the FGGA by a cross correlation of the high-pass filtered data to be 1.12 seconds and time shifted the data by this amount. Thus

$$CH_{4d,\text{vais}} = \frac{CH_{4w}}{(1 - mr_{\text{vais,ref}})}$$

and the flux

$$F_C = \overline{w'\rho'_{\text{CH}_4\text{d,vais}}} \tag{19}$$

The ratio $F_C/F$ turns out to be $1.080\pm0.047$, somewhat larger than method **B** mostly due to the apparently insufficiently accurate time shift procedure. However, this method had to be used for the 2012 data (e.g. Kohnert et al. (2017)) to enable wavelet decomposition.

**D** no correction for water vapour

$$F_D = \overline{w'\rho'_{\text{CH}_4w}} \tag{20}$$

The ratio $F_D/F$ is $0.793\pm0.093$. Thus, for our situation of methane emissions from arctic tundra the water vapour fluctuations lead to a flux under-estimation of 20% if not accounted for.

Figure 10 shows the above described for each horizontal flight section of the 2013 AirMeth campaign. We conclude, that even with a non-perfect humidity flux measurement, the dry mole fraction flux can be determined in polar regions with reasonable accuracy, in our case of the 2012 campaign an over-estimation of 4%.

## 5 Conclusions

We showed that aircraft are well suited tools to study methane emissions from Arctic tundra. The vertical fluxes of the most important greenhouse gases can be measured during low level flight legs with sufficient accuracy. We showed that a calibration




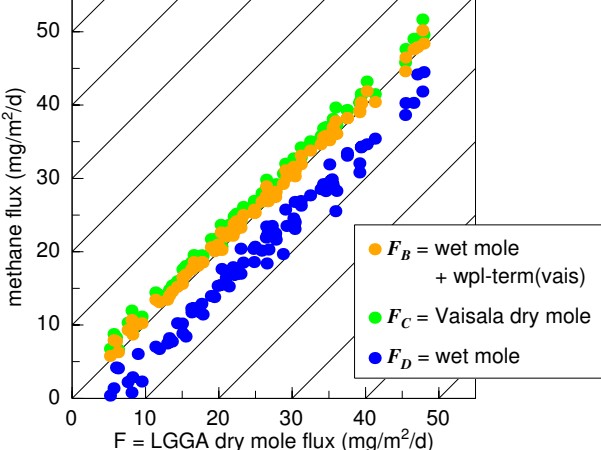

**Figure 10.** Comparison of different methods of accounting for humidity fluctuations in estimating methane flux from wet mole fraction measurements. The abscissa is the dry mole flux, $F$, Equation (16). Dark yellow is $F_B$, the wet mole flux plus WPL-term based on the Vaisala data according to Equation (18). Green represents $F_C$, Equation (19) and medium blue is the uncorrected wet mole flux, $F_D$, Eq.(20).

of the essential coefficients of an aicraft turbulence equipment can be achieved with high accuracy by exploiting suitably arranged flux measurement legs. The natural variations in parameters (airspeed, pitch) due to manually controlled flights are sufficient. The horizontal wind components are measured with an accuracy better than 0.2m/s during level flight legs. The level of white noise of the trace gas analyser does not allow to resolve the inertial subrange of turbulent fluctuations of $CO_2$ and $CH_4$

with sufficient accuracy. However, since the noise is uncorrelated with the vertical wind velocity, the cospectra show a -7/3 drop-off if sufficient data are available for averaging. We found the detection limit of the methane flux to be about $4\,\text{mg/m}^2\text{/d}$ and that of carbon dioxide to be about $1.4\,\text{g/m}^2\text{/d}$.

*Acknowledgements.* We thank the pilots, mechanics and engineers for their support. We thank Matthias Cremer for helpful discussions. The AirMeth campaigns were fully funded by Alfred Wegener Institute Helmholtz Centre for Polar and Marine Research. This work has

received funding from the Helmholtz Association of German Research Centres through a Helmholtz Young Investigators Group grant to T.S. (Grant VH-NG-821), and is a contribution to the European Union's Horizon 2020 research and innovation programme under grant agreement No. 727890, as well as to the Helmholtz Climate Initiative (REKLIM - Regional Climate Change). The National Ecological Observatory Network is a project sponsored by the National Science Foundation and managed under cooperative agreement by Battelle Ecology, Inc. This material is based upon work supported by the National Science Foundation [grant DBI-0752017]. Any opinions, findings, and conclusions

or recommendations expressed in this material are those of the author and do not necessarily reflect the views of the National Science Foundation.





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
