# Peer review of "New calibration procedures for airborne turbulence measurements and accuracy of the methane fluxes during the AirMeth campaigns"

_Atmospheric Measurement Techniques, 2017_

## Referee Comment (RC1) · Anonymous Referee #1 · 22 Feb 2018

[11pt]article

**Review: "The Polar 5 airborne measurement of turbulence and methane fluxes during the AirMeth campaigns:" Jörg Hartmann, Martin Gehrmann, Torsten Sachs, Katrin Kohnert, and Stefan Metzger**

2018 February 21

**1   General Comments**

This paper presents a potentially valuable approach to in-flight calibration and accuracy assessment for an airborne system that measures air-surface gas exchange from low altitude. This calibration and assessment procedure does not require the usual dedicated calibration flights, which consume at least a hour of flight time for each campaign flown. Instead multiple (10 or more) special dual-purpose flights are made both measuring the flux and providing calibration information. These consist of double passes, each covering the same ground track, but in opposite directions. Additional flights can also be flown suitable to other purposes and drawing on the calibration results from the dual-purpose flights. These calibration flights and procedures would seem to be appropriate to low-altitude flux fights. They would not directly generalize, for exam-

ple, to flights measuring mesoscale divergence (Lenschow et al., 2007), which may be required in the same campaign if significant mesoscale structure is present.

The language of the manuscript is excellent, apart from a number of minor editorial slips. Those that were caught are identified among the Technical Comments. The instrumentation and the flight tracks, including vertical profiles, are quite adequate for airborne flux measurement. With appropriate answers to several questions raised in the Specific Comments section, this paper is well suited to publication in *Atmospheric Measurement Techniques*.

**2 Specific Comments**

**2.0.1 Title**

The title more suggests a description of the turbulence and gas fluxes than a new technique of their calibration and assessment. One might onsider something like: "New calibration procedures not requiring dedicated calibration flights for airborne measurement of air-surface exchange developed using the Polar 5 Aircraft during the AirMeth campaigns."

Since Table 1 concerns itself primarily with the differences between outbound and inbound legs. It would be helpful to have be a separate table in the same format presenting absolute quantities that define the environment of these flights. Some of these already appear in Table 1, but would better fit in this new table. Such quantities include elapsed time to cover the pair of flight legs, the track direction $\chi_1$ the wind direction, the track length, the magnitudes of $v_\parallel$ and $v_\perp$ and possibly others.

Is the $\Delta t$ column meant to give the difference in travel time between the out and return legs, or is it to give the total elapsed time in traversing both legs: is it $t_{leg2} - t_{leg1}$ or is

it $t_{leg2} + t_{leg1}$? Later discussion (static pressure precision) suggests it is the latter, but also presents it as a function of position on the track, not a single number as given in Table 1. This could use some clarification in the table caption and text.

**2.0.2 Flight altitude**

The airspeed of 60 m s$^{-1}$ was given, but there was only one mention of the height above ground. That was 50 m above ground for the discussion of Figure 7. Since the ability to attribute flux measurements to surface characteristics deteriorates with height above ground, this parameter is important and could be included in the recommended (absolute environment) companion to Table 1.

**2.0.3 Calibration procedures**

**2.0.4 True Airspeed**

The primary concern is a lack of clarity in the development of Manuscript Equation (2) for the "Reference ground speed." The point of Manuscript Equation (2) appears to provide a determination of the true airspeed from the GPS/INS independent of the gust probe's measurements under conditions of the special dual-purpose flights. Some clarification would be helpful:

Quantities $v_{gi}$, $\chi_i$, and $\psi_i, i = 1, 2$ are probably averages of ground speed (magnitude) over their respective tracks (out and back). This should be made explicit. Presumably, the aircraft is on an autopilot rule to maintain airspeed (but not heading) and ground track (but not groundspeed). If so, however, the origin of Equation (2) is not readily discerned. The following assumptions appear to apply given the description of the reverse-track flights:

1. Wind velocity (magnitude and direction) does not change during the reverse-track maneuver.

2. True airspeed (but not heading) is held as near constant as possible, e.g. 60 m s$^{-1}$ (by autopilot or by human pilot)

3. Ground-track direction (but not ground-speed magnitude) is defined by a line segment on the surface, which is followed by the aircraft's autopilot (or human pilot) guided by GPS.

4. Averages are taken of airspeed, groundspeed, and the angle $\gamma_i = \chi_i - \psi_i$ between ground-track direction and aircraft heading for both legs.

Evaluating the "wind triangle" $\mathbf{V} = \mathbf{V_g} - \mathbf{V_{TAS}}$ (Manuscript Equation 1) for each of the two passes over the ground track is possible using the law of cosines:

$$V_i^2 = V_{gi}^2 + V_{TASi}^2 - 2V_{gi}V_{TASi}\cos\gamma_i \tag{1}$$

where the non-bold characters represent the magnitudes of the bold vectors, and all quantities are understood to be averages over their respective ground tracks ($i = 1, 2$). Since wind does not change ($\mathbf{V} = \mathbf{V_1} = \mathbf{V_2}$) the righthand sides of Equation (1) above for $i = 1, 2$ can be equated, eliminating windspeed as a variable. All other quantities are known from GPS/INS except for $V_{TAS}, i = 1, 2$. But the airspeed was held near constant allowing the assumption $V_{TAS1} = V_{TAS2} = V_{TASr}$ where $V_{TASr}$ is the reference that should be equal to $\overline{v_g}$ of the Manuscript Equation (2). Solving for $V_{TASr}$ one gets (assuming the algebra was correct)

$$V_{TASr} = \frac{V_{g1}^2 - V_{g2}^2}{2(V_{g1}cos\gamma_1 - V_{g2}cos\gamma_2)} \tag{2}$$

Equation (2) above bears some resemblance to Manuscript Equation (2), but time did not permit reconciling these two. They appear to have incompatible forms suggesting

that the authors used a different development to arrive at the manuscript's Equation (2). Some additional discussion of the assumptions and derivations actually used, in supplementary material if necessary, needs to be given.

Regarding Equation (5) (Page 6, lines 24, 25), is it appropriate to assume that the total pressure (static plus dynamic) is measured without error by the Rosemount probe as this statement appears to imply?

Using repeat instances (15 in this case) of reverse-track pairs to characterize the uncertainty in the assumptions (constant wind and TAS over the whole round trip) used to compute the correction factor for the dynamic pressure appears reasonable, and beneficial.

**2.0.5 Angle of attack**

These are promising procedures for determining the offset and slope of the "true" attack angle with respect to the ratio $q_\alpha/q_i$. First, a dedicated flight similar to that used by Crawford et al. (1996) but analyzed differently, provided what appears to be a clean calibration. Then the whole set of flights in the expedition was compounded to provide another estimate of the calibration parameters. With this large sample a restriction to those measurements reporting the same narrow range of vertical aircraft speed, vertical wind speed, and roll angle used with the earlier dedicated flights yielded a sample large enough to provide very nearly the same values for the calibration parameters $\alpha_0$ and $c_\alpha$.

**2.0.6 Angle of sideslip and Static Pressure Precision**

As with the angle of attack, these are straightforward and promising approaches.
**2.0.7 Accuracy of horizontal wind measurement**

The $v_\parallel$ is declared on page 12, line 10 to dominate "by far" the vector wind compared to the $V_\perp$ component. "By far" should be quantified. Apparently the flight legs were flown as much as possible parallel to the wind, but if that was clearly stated somewhere, I missed it.

**2.0.8 Methane Analyzer**

No comments: method looks appropriate.

**2.0.9 Accuracy of methane flux measurements**

The precision estimates for the methane flux use a technique described by reference to other publications. I had not seen it before. It looks intriguing. It would help the moderately interested reader (who can't justify digging through the references) to have a summary of the method. It's not intuitive how one gets a variance of noise error from a cross covariance input. Nor is it described how one finds the standard deviation over the blue-shaded areas. At the very least, the symbols $C_{11}$ and $p$ could be defined with indication of how to compute them. Perhaps $C_{11}$ is the autocovariance of the methane signal with itself and $p$ is the lag?

**2.0.10 Spectral analysis**

No comments: looks good

**2.0.11 Dry mole fraction flux**

Because the methane instrument and the water instrument did not share the same cell in the first two years, it was necessary to use different versions of the WPL terms.

The approach looks sound, but the notation suggests some possible problems, hopefully more apparent than real. Page 17, equation (16): The usual expression from WPL in the notation of this manuscript is $\overline{(w\rho_a)'CH'_{4d}}$, where $\rho_g$ is the density of the fraction of "dry" air. This computes the molar flux of $CH_4$ as the average of the product of the following departure quantities: the molar flux of dry air (as departure quantity $(w\rho_a)'$) times the dry-air mixing ratio of methane (as departure quantity $CH'_{4d}$). If $\rho'_{CH4d}$ is intended to be defined as $\rho'_a CH'_{4d}$ then it does not separate out the dry-air mass flux $(w\rho_a)'$ which is inconsistent with the method of WPL.

Otherwise this section is an informative exploration of the significance of the WPL correction for methane flux in the arctic and an effective demonstration of the effect on the uncertainty when different sensors for water vapor and methane must be used.

**3 Technical Corrections**

Mention is made of "precision" on several occasions, *e.g.*, page 4 and especially page 11 and following. Is "accuracy" a better term for some of these? Is the instrument or approach to be considered accurate at least to that stated precision (although its display may resolve greater precision)?

Page 6 Line 29: mispelled word: *tested* also page 11, Line 16.

Page 8 Line 6: "a plane with a fixed aerofoil" appears better as "an aircraft with a fixed aerofoil"

Page 12 line 7 to 8: "Rotating the wind components *into* an along-track..."

Page 13 lines 24,25: heat fluxes labeled in W m$^{-2}$ s$^{-1}$ instead of W m$^{-2}$

Page 14, Figure 6: caption line 3: "deviation has *been* calculated"

Page 15, Line 15: sentence fragment: needs a verb. Could say "Again, more scatter *is seen* for the trace gases than...."

Page 16, Line 4: consider recasting the sentence, e.g., "...the latter need to be taken into account in computing a mass flux from the measured (wet) mole fractions (Webb et al., 1980)."

**4  Reference Cited Above**

Lenschow, D.H., V. Savic-Jovcic, and B. Stevens, 2007: Divergence and vorticity from aircraft air motion measurements. *Journal of Atmospheric and Oceanic Technology*, **24(12)**, pp.2062–2072. Others cited above are listed in the manuscript itself.

---

## Referee Comment (RC2) · Anonymous Referee #2 · 17 May 2018

1. General comments This paper is a valuable contribution to the scientific scope of AMT. It shows a new method to perform a calibration of inflight data without time consuming flight hours for in-flight calibration. Especially as these calibration flights require perfect weather conditions it might be a time consuming and cost intensive phase within a measurement campaign. The calibration method describes in this paper show an alternative method to calibrate data from the air-data probes without specific flights. This method is based in principle on numerous flights and therefore is a kind of statistical approach. All equations show a "-" instead of "=". This is quite confusing and should be addressed by the author or editor. It might just be a problem of creating the pdf.

2. Instrumentation The description of the instrumentation and the aircraft is detailed and easily understandable. All relevant sensors and technical data are given.

3. Calibration of TAS The assumption of the wind changing by less than 0.25 m/s should be addressed more detailed. Which time period is considered? It is unlikely that the wind remains constant over the leg distance of the more than 150 km (first data in table 1). The derivation of equation (2) is missing. I understand this value not as ground speed but corrected speed in aircraft longitudinal direction. So the label of the value v g is confusing. It might be easier to perform an addition of the two vectors V g and V (as in equation (1)) to get the reference speed for the TAS calibration. The accuracy of this method is highly depending on the constancy of the wind. A constant change during the two legs seems to stay undetected as this could not be found in the differences of mean values in table 1. Each leg should be analysed separately with respect to time. A standard deviation per leg would be helpful to address this point. Equation (5) implies that the total pressure is measured correctly by the 5-hole probe and the error is only occurs in the static port of this probe. This is not a valid assumption for this kind of probe unless the flow angle at the probe is zero. As the flow angle at the probe is not mentioned a typical calibration curve of the 5-hole probe should be taken into account. The requirements to speed constancy are not mentioned at all. In principle the wind measurement should be independent of TAS but problems might arise by the fact that two legs are averages separately. What happens if one leg is flown at a different speed? The authors should address this point. It is not mentioned whether the computed values for wind and their differences in table 1 are obtained before calibration or thereafter.

The major question is the constancy of the wind during the whole roundtrip. What is the influence of a change in the wind over time and how can it be detected and eliminated?

4. Angle of attack calibration The method of angle of attack calibration is described in detail with sufficient explanations. The results are good especially as the flight conditions at low level over open sea are ideal. The comparison with the second method is
very helpful und shows the effectiveness of both approaches.

5. Angel of sideslip calibration The derivation of equation (11) is missing. For the sideslip angle calibration the same principle problem occurs as for the TAS calibration: a change of wind and / or TAS over time. An increased wind on one leg will lead to an increased residual error of the sideslip angle. This problem cannot be solved by this method unless the wind and TAS remain constant.

6. Static pressure precision The assessment of static pressure precision can only refer to a relative accuracy of the measurement. This is not addressed clearly. It is an interesting approach based on statistical methods.

---

## Author Comment (AC1) · 14 Jun 2018

We thank reviewer 1 for the review and for the helpful and constructive comments. We took nearly all of them into account in preparing a revised version of our manuscript. The specific comments are answered in the following:

Specific Comments:

*RC1: 2.0.1 Title*

[Figure]

*The title more suggests a description of the turbulence and gas fluxes than a new technique of their calibration and assessment. One might onsider something like: "New calibration procedures not requiring dedicated calibration flights for airborne measurement of air-surface exchange developed using the Polar 5 Aircraft during the AirMeth campaigns."*

We agree and changed the title to:

New calibration procedures for airborne turbulence measurements and accuracy of the methane fluxes during the AirMeth campaigns

*RC1: Since Table 1 concerns itself primarily with the differences between outbound and inbound legs. It would be helpful to have be a separate table in the same format presenting absolute quantities that define the environment of these flights. Some of these already appear in Table 1, but would better fit in this new table. Such quantities include elapsed time to cover the pair of flight legs, the track direction $\chi_1$ the wind direction, the track length, the magnitudes of $v_\parallel$ and $v_\perp$ and possibly others.*

We added a further table (Table A1) in the appendix listing all suggested quantities for each individual flight leg.

*RC1: Is the $\Delta t$ column meant to give the difference in travel time between the out and return legs, or is it to give the total elapsed time in traversing both legs: is it $t_{leg2} - t_{leg1}$ or is it $t_{leg2} + t_{leg1}$ ? Later discussion (static pressure precision) suggests it is the latter, but also presents it as a function of position on the track, not a single number as given in Table 1. This could use some clarification in the table caption and text.*

In Table 1 the difference between the mean time of each leg is given. The symbol $\Delta t$ was misleading, we changed to $\Delta \bar{t}$. This is now clarified in the caption of Table 1. The time duration needed to fly each leg is now also listed in Tabel A1. In the discussion

of the static pressure precision the symbol $\Delta t$ denotes the (position dependent) time difference.

*RC1: 2.0.2 Flight altitude*

*The airspeed of 60 m s$^{-1}$ was given, but there was only one mention of the height above ground. That was 50 m above ground for the discussion of Figure 7. Since the ability to attribute flux measurements to surface characteristics deteriorates with height above ground, this parameter is important and could be included in the recommended (absolute environment) companion to Table 1.*

The low level flights for flux measurements were mostly done at a height of 50 m above ground. We included the averaged height now in the new table A1. For the calibration of the turbulence probe, however, the height has little relevance and we therefore included in that section also some flight legs at greater heights. Those were actually flown to calibrate remote sensing instruments that are not subject of this paper. For the flux analysis, as e.g. presented in the papers of Kohnert et al. (2018) and Serafimovic, et al. (2018), many more legs were used that did not have an immediate successor on the return track.

*RC1: 2.0.4 True Airspeed*

*The primary concern is a lack of clarity in the development of Manuscript Equation (2) for the "Reference ground speed." The point of Manuscript Equation (2) appears to provide a determination of the true airspeed from the GPS/INS independent of the gust probe's measurements under conditions of the special dual-purpose flights. Some clarification would be helpful:*

*Quantities $v_{gi}$, $\chi_i$, $i = 1, 2$ are probably averages of ground speed (magnitude) over their respective tracks (out and back). This should be made explicit. Presumably, the aircraft is on an autopilot rule to maintain airspeed (but not heading) and ground*

*track (but not groundspeed). If so, however, the origin of Equation (2) is not readily discerned. The following assumptions appear to apply given the description of the reverse-track flights:*

*1. Wind velocity (magnitude and direction) does not change during the reverse-track maneuver.*

*2. True airspeed (but not heading) is held as near constant as possible, e.g. 60 m s$^{-1}$ (by autopilot or by human pilot)*

*3. Ground-track direction (but not ground-speed magnitude) is defined by a line segment on the surface, which is followed by the aircraft's autopilot (or human pilot) guided by GPS.*

*4. Averages are taken of airspeed, groundspeed, and the angle $\gamma_i = \chi_i - \Phi_i$ between ground-track direction and aircraft heading for both legs.*

*Evaluating the "wind triangle" $V = V_g - V_{\text{TAS}}$ (Manuscript Equation 1) for each of the two passes over the ground track is possible using the law of cosines:*

$$V_i^2 = V_{gi}^2 + V_{TASi}^2 - 2V_{gi}V_{TASi}\cos\gamma_i \tag{1}$$

*where the non-bold characters represent the magnitudes of the bold vectors, and all quantities are understood to be averages over their respective ground tracks (i = 1, 2). Since wind does not change $(V = V_1 = V_2)$ the righthand sides of Equation (1) above for i = 1, 2 can be equated, eliminating windspeed as a variable. All other quantities are known from GPS/INS except for $V_{TAS}i, i = 1, 2$. But the airspeed was held near constant allowing the assumption $V_{TAS1} = V_{TAS2} = V_{TASr}$ where $V_{TASr}$ is the reference that should be equal to $\overline{v_g}$ of the Manuscript Equation (2). Solving for $V_{TASr}$ one gets (assuming the algebra was correct)*

$$V_{TASr} = \frac{V_{g1}^2 - V_{g2}^2}{2(V_{g1}\cos\gamma_1 - V_{g2}\cos\gamma_2)} \tag{2}$$

*Equation (2) above bears some resemblance to Manuscript Equation (2), but time did not permit reconciling these two. They appear to have incompatible forms suggesting that the authors used a different development to arrive at the manuscript's Equation (2). Some additional discussion of the assumptions and derivations actually used, in supplementary material if necessary, needs to be given.*

We thank the reviewer for pointing out the deficiency of the manuscript in this part. We clarified our assumptions and rephrased the entire derivation of the reference true airspeed. Please refer to the revised manuscript.

Though mathematically correct, Equation (2) of the reviewer leads to problems in the practical use, as a relatively small difference of two larger quantities appears in the denominator. Assymmetries in the measurement inaccuracies then cause large scatter of the result. The ground speed appearing the the square in the numerator is available with a high accuracy by the gps. The true heading direction, however, is less accurate and represents actually the largest uncertainty in the entire wind derivation. Thus, the small difference of the terms involving direction in the denominator leads to large scatter.

*RC1: 2.0.5 Angle of attack ...*

No action required.

*RC1: 2.0.6 Angle of sideslip and Static Pressure Precision ...*

No action required.

*RC1: 2.0.7 Accuracy of horizontal wind measurement*

*The $v_\parallel$ is declared on page 12, line 10 to dominate "by far" the vector wind compared to the $V_\perp$ component. "By far" should be quantified. Apparently the flight legs were flown*

*as much as possible parallel to the wind, but if that was clearly stated somewhere, I missed it.*

The formulation was misleading, we rephrased that sentence. We meant an error propagation for the along track component only, as in this component the uncertainty of the dynamic pressure is the major contributer. An error assessment for both wind components is given further down in that section.

*RC1: 2.0.8 Methane Analyzer*

No action required.

*RC1: 2.0.9 Accuracy of methane flux measurements*

*The precision estimates for the methane flux use a technique described by reference to other publications. I had not seen it before. It looks intriguing. It would help the moderately interested reader (who can't justify digging through the references) to have a summary of the method. I's not intuitive how one gets a variance of noise error from a cross covariance input. Nor is it described how one finds the standard deviation over the blue-shaded areas. At the very least, the symbols $C_{11}$ and p could be defined with indication of how to compute them. Perhaps $C_{11}$ is the autocovariance of the methane signal with itself and $p$ is the lag?*

We added in the revised manuscript brief summaries of the methods cited and used to assess the instrumental noise and the flux detection limits. Further, we added the missing explanations of the symbols and how to calculate the standard derivation that is now marked in the figure (was Figure 6, now Figure 7).

*RC1: 2.0.11 Dry mole fraction flux*

*Because the methane instrument and the water instrument did not share the same cell in the first two years, it was necessary to use different versions of the WPL terms. The approach looks sound, but the notation suggests some possible problems, hopefully more apparent than real. Page 17, equation (16): The usual expression from WPL in the notation of this manuscript is $\overline{(w\rho_a)' CH'_{4d}}$, where $\rho_g$ is the density of the fraction of "dry" air. This computes the molar flux of $CH_4$ as the average of the product of the following departure quantities: the molar flux of dry air (as departure quantity $(w\rho_a)'$) times the dry-air mixing ratio of methane (as departure quantity $CH'_{4d}$). If $\rho'_{CH4d}$ is intended to be defined as $\rho'_a CH'_{4d}$ then it does not separate out the dry-air mass flux $(w\rho_a)'$ which is inconsistent with the method of WPL. Otherwise this section is an informative exploration of the significance of the WPL correction for methane flux in the arctic and an effective demonstration of the effect on the uncertainty when different sensors for water vapor and methane must be used.*

The reviewer pointed to a slight inconsistency in the notation. We intended $\rho_{CH4d}$ to be defined as the density of methane. We now dropped the subscript $d$ in the revised manuscript to be consistent with the following formulas.

The technical corrections have all been applied.

---

## Author Comment (AC2) · 14 Jun 2018

We thank reviewer 2 for the review and for the helpful and constructive comments. We took nearly all of them into account in preparing a revised version of our manuscript. The specific comments are answered in the following:

*RC2: 1. General comments .... All equations show a "-" instead of "=".*

The files uploaded to the copernicus site passed the copernicus validation checks. When downloading the pdf, all equations are printed correctly. We cannot reproduce

that effect. What kind of pdf-viewer did the reviewer use ? The technical editor should look into that issue.

*RC2: 2. Instrumentation*

No action required.

*RC2: 3. Calibration of TAS The assumption of the wind changing by less than 0.25 m/s should be addressed more detailed. Which time period is considered? It is unlikely that the wind remains constant over the leg distance of the more than 150 km (first data in table 1). The derivation of equation (2) is missing. I understand this value not as ground speed but corrected speed in aircraft longitudinal direction. So the label of the value $v_g$ is confusing. It might be easier to perform an addition of the two vectors $V_g$ and $V$ (as in equation (1)) to get the reference speed for the TAS calibration. The accuracy of this method is highly depending on the constancy of the wind. A constant change during the two legs seems to stay undetected as this could not be found in the differences of mean values in table 1. Each leg should be analysed separately with respect to time. A standard deviation per leg would be helpful to address this point. Equation (5) implies that the total pressure is measured correctly by the 5-hole probe and the error is only occurs in the static port of this probe. This is not a valid assumption for this kind of probe unless the flow angle at the probe is zero. As the flow angle at the probe is not mentioned a typical calibration curve of the 5-hole probe should be taken into account. The requirements to speed constancy are not mentioned at all. In principle the wind measurement should be independent of TAS but problems might arise by the fact that two legs are averages separately. What happens if one leg is flown at a different speed? The authors should address this point. It is not mentioned whether the computed values for wind and their differences in table 1 are obtained before calibration or thereafter. The major question is the constancy of the wind during the whole roundtrip. What is the influence of a change in the wind over time and how*

*can it be detected and eliminated?*

The calibration method does not require a wind changing less than 0.25 m/s for each of the out- and return-flight manoeuvres. We rather argue that with the multitude of such manoeuvres possible wind changes between out- and return flight are randomly distributed and their influence on the eventual calibration parameter is considerably reduced due to the averaging process.

We now more explicitly show the derivation of Equation (2) and rephrased the entire derivation of the reference true airspeed. Please refer to the revised manuscript. We also changed several symbols especially that referring to the reference true airspeed.

The accuracy of this method ist not highly dependent on the wind being constant for each individual pair of return track manoeuvres, as we use a large number of those manoeuvres to find the calibration parameters. There are some manoeuvres (e.g. #1, #4 and #14, we added a sequential numbering in table 1 in the revised version of the manuscript) where the wind changes by about -0.6 m/s or -0.8 m/s between out and return flight. Most manoeuvres, however, have a wind change of about $\pm 0.2$ m/s and the average of all changes is -0.11 m/s.

The reviewer correctly pointed out a deficiency in our correction of the static pressure measurements. We now include the probe's error as a function of probe angle as found by wind tunnel tests of an identical model. For most situations of level flights this correction term, however, is very small.

The out- and return flights have been flown at the same manually controlled airspeed. The actual differences in airspeed between both legs are very small. We now include the true airspeed in Table 2, the list of parameters for each separate flight leg.

The computed values listed in Table 1 (and also those in the new Table 2) are calculated after the calibration. This is now mentioned in the table headings.

*RC2: 4. Angle of attack calibration The method of angle of attack calibration is described in detail with sufficient explanations. The results are good especially as the flight condi- tions at low level over open sea are ideal. The comparison with the second method is very helpful und shows the effectiveness of both approaches.*

No action required.

*RC2: 5. Angel of sidesliple calibration The derivation of equation (11) is missing. For the sideslip angle calibration the same principle problem occurs as for the TAS calibration: a change of wind and / or TAS over time. An increased wind on one leg will lead to an increased residual error of the sideslip angle. This problem cannot be solved by this method unless the wind and TAS remain constant.*

We added further to the derivation of Equation (11) (now Equation 13). The reviewer correctly pointed out, that for a single pair of out- and retur-flights no distinction can be made between a change of wind and a possible misalignment of the probe. However, with a large number of return manoeuvres in different situation and on different days we can assume that possible wind changes are randomly distributed, and thus wind contribution to the average of all residuals (the beta misalignments) should vanish. We further explained this in the revised version of the manuscript.

*RC2: 6. Static pressure precision The assessment of static pressure precision can only refer to a relative accuracy of the measurement. This is not addressed clearly. It is an interesting approach based on statistical methods.*

This is true. Offset errors in the static pressure cannot be detected by this method. We added this comment in the revised manuscript.